# Salivary Adiponectin and Albumin Levels on the Gingival Conditions of Patients Undergoing Bariatric Surgery: A Cohort Study

**DOI:** 10.3390/ijerph20075261

**Published:** 2023-03-24

**Authors:** Silvia Helena de Carvalho Sales-Peres, Jack Houghton, Gabriela de Figueiredo Meira, Patrícia Garcia de Moura-Grec, Sergio Luis Aparecido Brienze, Belkais Abuuasha Karim, Guy Howard Carpenter

**Affiliations:** 1Department of Pediatric Dentistry, Orthodontics and Public Health, Bauru School of Dentistry, University of São Paulo, Bauru 17012-901, Brazil; 2Department of Surgery, Faculty of Medicine of Sao Jose do Rio Preto, São José do Rio Preto 15090-000, Brazil; 3Center for Host-Microbiome Interactions, Faculty of Dental, Oral and Craniofacial Sciences, King’s College London, London WC2R 2LS, UK

**Keywords:** salivary biomarkers, periodontal disease, obesity, adiponectin, albumin

## Abstract

This study analyzed the salivary proteomics, adiponectin and albumin, related to weight loss and periodontitis in patients undergoing bariatric surgery. This study included fourteen patients with morbid obesity (body mass index, BMI > 40 kg/m^2^) who underwent bariatric surgery Roux-en-Y gastric bypass (RYGB) in System Health Public in Brazil. Data on demographic and anthropometric measures were extracted from medical records preoperatively and 6 and 12 months post-surgery. The variables assessed were: probing pocket depth (PPD), clinical attachment loss (CAL), bleeding on probing (BOP), and stimulated whole-mouth saliva. In this study, saliva samples were analyzed by electrophoresis and immunoblotting. The ELISA kit was used to measure the MMP8 levels to determine potential markers for obesity. Adiponectin and albumin levels were also evaluated. Weight loss was associated with significant changes in patients’ periodontal clinical data. Although 7 out of 10 periodontal patients showed an increase in salivary adiponectin levels after root planning treatment, when analyzed by Western blotting, the increase was not statistically significant (21.1 ± 4.8 to 26.3 ± 9.4 arbitrary units, *p* > 0.99). There was no correlation between albumin levels and salivary adiponectin pre-surgery, nor 6 months or 12 months after surgery. Weight loss was not improved by low-grade inflammation in bariatric patients, since albumin levels were similar between periods. Periodontitis is an inflammatory disease that is modulated by several factors, among which adiponectin plays an important role for the treatment of periodontal disease.

## 1. Introduction

Obesity is a chronic disease that has increased substantially in recent years and predictions suggest that 58% of adults worldwide will be overweight or obese by 2030 [1]. Morbid obesity has been associated with increased comorbidities such as coronary artery disease, diabetes, and destructive periodontal disease [1,2]. Effective treatment for morbid obesity includes gastrointestinal surgery and such interventions have increased with the rise of the disease epidemic. However, there are a number of complications that accompany bariatric surgeries, some of which manifest in the oral cavity [1,3].

In patients with type 2 diabetes mellitus, metabolic syndrome and coronary artery disease, reduced plasma levels of adiponectin may play a key role in the development of insulin resistance. Although the mechanisms underlying the anti-inflammatory properties of adiponectin are not well understood, the anti-atherogenic and anti-inflammatory properties of adiponectin may be related in part to its ability to stimulate nitric oxide production by the vascular endothelium [4]. Adiponectin is a 30 kDa adipokine originally thought to be exclusively secreted by adipocytes, but more recent studies have shown that it is also secreted by salivary gland epithelial cells [5], osteoblasts [6] and cardiomyocytes [7]. Adiponectin circulates in the blood in low (LMW), medium (MMW) and high molecular weight (HMW) forms; the levels of each are controlled by adipocytes [8]. It has roles involved in the development of insulin resistance [9], inflammation [10], inhibition of murine osteoclastogenesis and bone resorption [11]. It is suggested that plasma levels of adiponectin be used as a biomarker for obesity [12] and there are few studies on the nature of adiponectin in saliva; a study by Toda et al. was the first to measure adiponectin in saliva after Groschl et al. suggested its existence [13,14].

In this sense, saliva plays a key role in protecting and maintaining the integrity of the oral mucosa through lubrication, buffering action, antibacterial and antiviral activity, and food digestion [15]. Proteomic studies have identified and characterized over 1166 salivary proteins and peptides [16]. Most of these are commonly found in plasma and some are produced and secreted exclusively by the salivary glands, having no correlation with blood levels [17]. Saliva can significantly contribute to disease screening, risk assessment, intervention evaluation, recurrence prediction, and other evaluations of prognostic outcomes [18]. The salivary protein biomarkers demonstrated that these contents highlight information for the detection of oral and systemic diseases [19].

Periodontal disease is a chronic disease characterized by pathogenic levels of periodontal pathogens, as a result of inflammatory and immune-mediated tissue changes [18]. The bacterial biofilm formation initiates gingival inflammation; however, periodontitis initiation and progression depend on dysbiotic ecological changes in the microbiome in response to nutrients from gingival inflammatory and tissue breakdown products that enrich some species and anti-bacterial mechanisms that attempt to contain the microbial challenge within the gingival sulcus area once inflammation has initiated [20].

Collagenase-2 (MMP8) is a tissue-degrading host proteinase secreted by neutrophils and fibroblasts. It has a role in maintaining a healthy periodontium [19] and is expressed at elevated levels in patients with periodontal disease [21]. MMP9 is one of the major collagen-degrading enzymes in saliva, which is associated with periodontitis [22] and MMP-13 has also been found to be involved in periodontal tissue destruction and alveolar bone resorption. In fact, MMP-8 (or collagenase 2) is currently one of the most promising biomarkers for periodontitis in oral fluids [23].

Therefore, the relationship between periodontal disease and obesity has been widely discussed in the literature [1,18]. Obesity provides an imbalance in the levels of pro-inflammatory cytokines, impairing the immune response of obese patients [10]. It has also been reported in the literature that the inflammatory process caused by obesity leads to a decrease in albumin and transferrin, leading to a change in oxygen diffusion, nutrient transport, migration of polymorphonuclear cells and monocytes/macrophages and the diffusion of antibodies. In present study, we found no correlation between salivary albumin levels pre-surgery, nor 6 months or 12 months after surgery. Contradictory results have been reported about the relationship between severe periodontitis with low albumin levels [24].

To date, little is known about the association between weight loss and adiponectin and albumin levels in periodontal patients. The hypothesis of this study was that weight loss would increase salivary adiponectin and albumin levels, which may impact the oral health of periodontal patients. Salivary levels of MMP8 were used as markers of gingival health.

## 2. Materials and Methods

### 2.1. Design and Participants

This is an observational, longitudinal study with morbidly obese patients who were under multidisciplinary follow-up to undergo bariatric surgery treated at the Brazilian Unified Health System (SUS). Obesity was assessed using the body mass index (BMI). Values were obtained by accessing weight in kilograms and height in meters and centimeters. Patients were classified as morbidly obese if they had a BMI from 40 kg/m^2^ [25].

In a previous study, it was demonstrated that anthropometry and DXA were equally useful in obese women, whereas anthropometry had no predictive power and DXA was the only acceptable predictor in obese men [26]. Our sample consisted mostly of women.

Patients were recruited from April 2010 to March 2012, treated at the School of Medicine Hospital, São José do Rio Preto, State of São Paulo, Brazil. Individuals with less than six teeth, smokers, pregnant women, carriers of infectious diseases and medicated with anti-inflammatories or antibiotics at least 3 months before this study were excluded. Therefore, from the total initial sample, 14 patients underwent Roux-en-Y gastric bypass (RYGB) bariatric surgery and were reassessed 6 and 12 months after surgery.

### 2.2. Ethical Aspects

This study was conducted in accordance with the Declaration of Helsinki, and was approved by the Ethics Committee for Research with Human Beings of the School of Medicine Hospital–FAMERP (process nº 0315/08). Written consent was obtained from all subjects prior to the start of this study, after being informed of the study objectives.

### 2.3. Data Collect

Complete medical histories were obtained from the patient’s records and confirmed by interview. Health information, C-reactive protein (CRP) and glucose levels were obtained from the patient’s medical record.

### 2.4. Oral Clinical Evaluation

One dentist who had been trained at the beginning of this study performed all the examinations (Kappa > 0.80) for intra-examiner concordance.

The measurements were performed with the use of a standard periodontal clinical probe 15 with a cylindrical shape, 15 mm scale, 1.75° cone taper, and 0.5 mm probe tip diameter (Quinelato, Schobell Industrial Ltda., Rio Claro, SP, Brazil) for the clinical examination of all present teeth. Periodontal condition measures including bleeding on probing (BOP), presence of calculus (PC), probing depth (PD), and clinical attachment loss (CAL) were assessed at baseline, month 6 and month 12 after bariatric surgery [27].

The gingival and calculus indices were used to measure the presence of bleeding or calculus, respectively. Bleeding on probing indicates the presence of gingivitis. PPD was measured as the distance between the gingival margin and the bottom of the gingival crevice. CAL was determined by measuring the distance between the cement–enamel junction and the bottom of the gingival crevice. Sites with PPD equal to or greater than 4 mm and with CAL equal to or greater than 3 mm constituted the presence of periodontitis [28]. All patients received professional prophylaxis at baseline.

Once samples were taken, all individuals underwent full-mouth scaling and root planning (SRP) and participants were given standard oral hygiene instructions. For daily dental hygiene the brushing technique used was Bass modified, with a soft brush, twice a day. With supragingival scaling and polishing, this significantly improved the gingival bleeding.

### 2.5. Saliva Collection and Storage

Saliva sample collection was performed at three different times, using a standardized saliva collection protocol. All participants were instructed not to ingest food or drink for at least 1 h before sample collection. Initially, they rinsed their mouths with 5 mL of potable deionized water, and then were asked to swallow saliva for 5 min. After this period, patients were instructed to chew a small sheet of parafilm and spit out all the saliva accumulated in the mouth into a plastic tube immersed in ice for 10 min (stimulated flow). When all saliva samples were obtained, they were thawed at room temperature, divided into 1.5 mL tubes, and centrifuged at 14,000× *g* at 4 °C for 15 min to remove all debris [29]. The supernatant was saved and transferred to new 1.5 mL tubes, and the samples were frozen at −80 °C until ready to use.

### 2.6. Electrophoresis and Immunoblotting

Saliva samples were prepared for electrophoresis by dilution in sample buffer of 4× concentration (lithium dodecyl sulfate, Life Technologies Ltda., Carlsbad, CA, USA)) with the addition of 0.5 M dithiothreitol (Sigma-Aldrich Corp., Milwaukee, WI, USA) to the sample buffer solution. Electrophoresis was performed according to the manufacturer’s instructions using 4–12% Bis-Tris gels (Life Technologies Ltda.) and MES running buffer. Separated proteins were electrotransferred to nitrocellulose membranes, stained with FITC and photographed under UV light (350 nm). The blots were probed with an affinity isolated antibody fraction of rabbit antiserum to a synthetic human adiponectin peptide corresponding to amino acid residues 225–244 (Sigma-Aldrich), an immunogen affinity purified fraction of antiserum for a synthetic peptide from the human MMP8 hinge region (Abcam, Waltham, MA, USA) or monoclonal anti-human serum albumin derived from HSA-11 hybridoma (Sigma). Binding was detected using an affinity purified goat-ant-rabbit IgG labeled with peroxidase (Dako, Ely, Cambridgeshire, UK) and then the Clarity TM Western ECL substrate detection system (Bio-Rad Laboratorios Brasil Ltda., Lagoa Santa, MG, Brazil). Chemiluminescence was detected by the ChemiDoc TM MP Imaging System (Bio-Rad Laboratorios Brasil Ltda., MG, Brazil) Molecular masses were determined by comparison with standard proteins.

### 2.7. Enzyme Immunosorbent Assay (ELISA)

MMP8 was included in this study to investigate the health of periodontium to relate the clinical evaluation. In this way, MMP8 levels in patients’ periodontal samples were measured by ELISA. A volume of 100 µL mouse anti-human MMP8 (R&D Systems) at 2.0 µL/mL diluted in PBS per well was incubated in a 96-well plate and incubated overnight at room temperature. Three washes were performed in phosphate-buffered saline with 0.05% tween (PBS-T). A volume of 300 µL of phosphate-buffered saline with 1% BSA was added to each well and allowed to incubate at room temperature for one hour, followed by three washes with PBS-T. A serial dilution of 100 µL of samples and standards was performed in duplicate and the plates incubated for 2 h at room temperature, followed by 3 washes with PBS-T. A volume of 100 µL of goat anti-human MMP8 (R&D Systems) at 50 ng/mL diluted in phosphate-buffered saline with 1% BSA was added to each well and allowed to incubate for 2 h at room temperature, followed by three washes with PBS-T. A volume of 100 µL of 1:200 streptavidin-HRP in phosphate-buffered saline with 1% BSA was added to each well and allowed to incubate for 20 min at room temperature, followed by three washes with PBS-T. A volume of 100 µL 1:1 H2O2 to Tetramethylbenzadine Substrate Solution added to each well and left to incubate for 20 min at room temperature followed by addition of 50 µL 2MH2SO4 Stopping Solution. Absorbance was read at 450 nm using a plate reader (BioRad, Hemel Hempsted, UK) [29].

### 2.8. Statistical Analysis

Descriptive statistics (mean, standard deviation, absolute and relative frequencies) were calculated after verifying the normal distribution (Shapiro–Wilk). The analysis included ANOVA for repeated measures to verify the difference between the three periods (preoperative and 6 and 12 months post-operatively) followed by Tukey’s test for post hoc comparison. Pearson correlations were performed to assess the correlation between adiponectin and albumin levels preoperatively, 6 and 12 months after surgery. The significance level was set at *p* < 0.05. The analyses were performed using the JAMOVI 2.2.5 software. The protein will be analyzed by Scaffold software.

## 3. Results

A decrease in the bariatric patient’s BMI from preoperative levels at 6 months, and a further decrease at 12 months, indicates a successful weight loss intervention. This weight loss was associated with significant changes in the patient’s periodontal clinical data, with both clinical attachment loss (CAL) and pocket depth (PD) showing an increase after 6 months, indicating a worsening periodontal condition. At 12 months, there is a reduction in CAL and PD suggesting a recovery of the periodontal condition (Table 1).

Clinical evaluation indicated a reduction in periodontal disease. Although gingival recession increased, pocket depth and bleeding on probing decreased after surgery (Table 2).

Nine out of ten bariatric patients showed a decrease in salivary adiponectin levels 6 months after surgery compared to the preoperative sample (Figure 1). This corresponded to the greatest decrease in body mass (Table 1) and worsening gingival health, as evidenced by clinical loss of attachment and pocket depth. Subsequently, six of the ten patients showed an increase in salivary adiponectin levels 12 months after surgery similar to preoperative levels (Figure 1). At that point, clinical periodontal measurements were better than at 6 months, but not the same as preoperative levels.

When normalized to the mean adiponectin intensity of pre-surgical samples, there was no change in adiponectin intensity (*p* = 0.084); however, when sample #37 was removed from the analysis, as it appeared to be an outlier, there was a significant decrease at 6 months after sampling (*p* < 0.05). There was an increase in levels from 6 months to 12 months, which were not different from preoperative levels (Figure 2).

To determine whether salivary adiponectin levels were affected by gingival health, the periodontal disease of non-obese patients was analyzed. Analysis of their saliva samples revealed that, although seven out of ten periodontal patients had an increase in salivary adiponectin levels, when analyzed by Western blotting (Figure 3), the increase was not statistically significant (21.1 ± 4.8 for 26.3 ± 9.4 arbitrary units, *p* > 0.99).

Salivary levels of MMP 8 were used to assess patients’ gingival health. Five out of ten periodontal patients showed a decrease in MMP8 salivary levels during the period when analyzed by Western blotting (results not shown), which were not statistically different (Figure 4).

However, after analysis of the same samples by ELISA, eight of the ten patients showed a decrease in salivary MMP 8 levels (Figure 5) and a significant reduction in mean MMP8 levels (*p* = 0.044). MMP8 analysis of saliva from bariatric patients was only possible using prior Western blots due to limited sample volumes. These results showed no difference in the preoperative period compared to the post-operative period (6 or 12 months).

No significant difference was found between the pre-surgery, 6 month post-surgery or 12 month post-surgery bariatric patient salivary albumin levels (*p* > 0.05) (Figure 6).

Salivary adiponectin in samples from bariatric patients were plotted against pre-surgery salivary albumin levels, either 6 months or 12 months after surgery, there was no significant correlation (R^2^ = 0.0456).

## 4. Discussion

This is the pioneering study to examine the association of weight loss and salivary albuminum and adiponectin with the interaction between bariatric surgery and periodontal disease. Our data highlighted that bariatric and periodontitis patients presenting similar albumin levels, indicating that weight loss was not improved by low-grade inflammation.

Adipose tissue is an active endocrine organ that secretes bioactive factors such as adipokines. A study investigated total levels of adiponectin and its oligomeric profile in saliva from obese subjects compared to age- and sex-matched control subjects. There were no statistical relevant differences of total salivary adiponectin and its oligomers in obese patients compared to healthy controls [30].

The different modulation of adiponectin in saliva and serum of obese patients could be explained assuming that different sources of the adipokines detected in saliva may originate from both local production or may be transported by other biological fluids [31]. Salivary determinations of adiponectin may contribute to the elucidation of the physiology and the role of this adipokinein various conditions, such as obesity, insulin resistance, reproduction, energy balance, and stress response and may represent a driving factor for insulin resistance, local inflammation in patients with obesity and obesity-related diseases [30]. In our study, Western blot analysis of salivary adiponectin in patients undergoing bariatric surgery showed a non-significant increase 6 months after surgery, followed by a non-significant decrease to approximately preoperative levels after 12 months. However, the 6-month data were skewed by a data point from patient 34, which is 16× the level of the next highest value and 155× the level of the lowest. If this outlier is removed, then a significant decrease in salivary adiponectin is observed (*p* = 0.0002) between pre-surgery salivary adiponectin levels and post-6 month levels and a non-significant increase (*p* = 0.2031) between the 6 month and 12 month levels. There were no clinical grounds for excluding this outlier as an anomaly, but its presence can be explained by the pre-surgical adiponectin level of the corresponding patient being almost too low. Weight loss was expected to lead to an increase in salivary adiponectin levels, as BMI levels show reduced obesity, which is correlated with higher adiponectin levels [32] and plasma levels of HMW adiponectin have been reported to increase in 40 ± 15% in 1 month and in 58 ± 8% at 12 months after gastric bypass surgery in one study [33] and from 11.4 ± 0.7 mg/L before surgery to 15.7 ± 0.7 and 19.8 ± 1.0 at 6 and 12 months after surgery, respectively [34]. However, a drop in adiponectin levels after surgery, followed by an increase back to preoperative levels, has been demonstrated earlier in plasma MMW adiponectin [35]. The return to preoperative levels could be explained by a rebalancing of the systemic adiponectin balance after surgery that did not stabilize after 6 months. This could be due to post-surgery systemic inflammation, dietary behavioral changes post-surgery, or the effect of bypass surgery on the gastric microbiota and these effects on the body as a whole.

Analysis of the clinical data shows a significant increase in pocket depth and clinical attachment level 6 months after surgery, followed by a decrease from 6 month to 12 months after surgery. The change in periodontal disease between baseline and 12 months was from 3.21 to 2.90 mm pocket depth, which is not clinical relevant. This suggests a worsening of the periodontal condition of patients undergoing Roux-en-Y gastric bypass surgery, followed by a return to a healthier periodontal condition, a negative correlation with salivary adiponectin levels.

A previous study highlighted that MMP9 was associated with obesity and periodontitis [22] and MMP-9 has been shown to be a more sensitive marker for periodontal inflammation. In this way, MMP-8 and MMP-9 are extremely valuable diagnostic tools in treating periodontitis, while MMP-13 affects the activity of osteoclasts and bone resorption, contributing to the destruction of the periodontal tissue [23]. The periodontal patient set was brought into the study to determine whether improvement of a diseased periodontium would affect salivary adiponectin levels, as it has previously been shown that an improvement in periodontal condition correlates with increases in adiponectin [36]. However, Western blot data do not suggest a correlation between gingival health and salivary adiponectin levels. The results suggest that salivary adiponectin levels may be associated with obesity-induced changes in the periodontium, as they closely followed clinical indices of gingival disease, while the biochemical marker MMP8 was less decisive. This is perhaps due to the difference we observed between MMP8 levels measured by immunoblotting and ELISA. Immunoblotting indicated a single band with the correct molecular weight, but it is only one antibody and non-specific binding cannot be excluded. ELISAs are generally considered more specific as they use two antibodies against the same protein. A future study may investigate whether the underlying pathophysiological mechanisms of metalloproteinases could lead to novel diagnostic and therapeutic methods.

Although hormones are known to be involved in albumin metabolism, no single hormone has been identified as a prominent factor in the physiological regulation of albumin metabolism. Albumin function is an indicator of nutritional status and actions of antioxidants. There is considerable knowledge about albumin synthesis, but the factors that regulate albumin degradation are not clearly established. The role of albumin itself in the synthesis of specific liver proteins is not fully understood, although in vitro data suggest that albumin may be a key factor. The apparent increase in the hepatic synthesis of specific proteins may be able to sustain not only colloid osmotic pressure, but also albumin transport and function [37].

Periodontal microbes that trigger an inflammatory response, such as T. denticola, increase the levels of salivary albumin. Additionally, these microbes use albumin, as well as immunoglobulins, as potential energy sources [22]. A recent systematic review analyzed albumin concentration and found that chronic kidney disease patients with periodontitis presented lower albumin levels, indicating that their health was at risk. The authors hypothesized that there might be an inverse relationship between periodontal disease and serum albumin concentration [38]. Thus, new studies should investigate renal and hepatic conditions after bariatric surgery, considering these hypotheses.

Elevated urinary albumin excretion independently and continuously predicts an increased risk of cardiovascular disease, even at levels below the threshold for the diagnosis of microalbuminuria. Albuminuria likely reflects systemic endothelial dysfunction rather than directly contributing to increased risk of cardiovascular disease [39]. However, reductions in albuminuria after bariatric surgery independently confer a lower risk of fatal cardiovascular disease [40].

Salivary protein and albumin were positively correlated with periodontal patients, which demonstrated that the increase in salivary protein concentration in individuals with gingivitis or periodontitis was caused by plasma protein leakage [41]. In our study, analysis of salivary albumin was performed to determine whether increases and decreases in adiponectin could be explained by varying levels of serum leakage. Mean salivary albumin levels remained constant after bariatric surgery. This suggests that the variation in salivary adiponectin is not a reflection of altered serum leakage in the oral cavity.

There were some limitations in the present study. First, the sample size was small; however, a causal relationship between weight loss and adiponectin levels of periodontal patients has been established. Second, the population of this study was the convenience sample (non-probabilistic sampling), which may have introduced some selection bias. Finally, there were limitations in the evaluation of saliva and plasma of albumin levels to compare between these biomarkers. However, there were several important strengths of our study that are also worth mentioning. This study examined the association of salivary albumin and adiponectin with the interaction between bariatric surgery and periodontal disease. In addition, important covariates were measured and controlled in the study analysis.

Adiponectin is produced by fat cells and is considered an anti-inflammatory molecule. There was no correlation between salivary adiponectin levels and salivary albumin levels, which suggests that salivary adiponectin does not enter saliva through the same pathway as albumin, but may reflect local production.

## 5. Conclusions

These findings provide insight that weight loss was not improved by low-grade inflammation in bariatric patients, since albumin levels were similar between study periods. Periodontitis is an inflammatory disease that is modulated by several factors, among which adiponectin plays an important role for the treatment of periodontal disease in candidates of obesity surgery, while the mechanisms and clinical application require further exploration.

## Figures and Tables

**Figure 1 ijerph-20-05261-f001:**
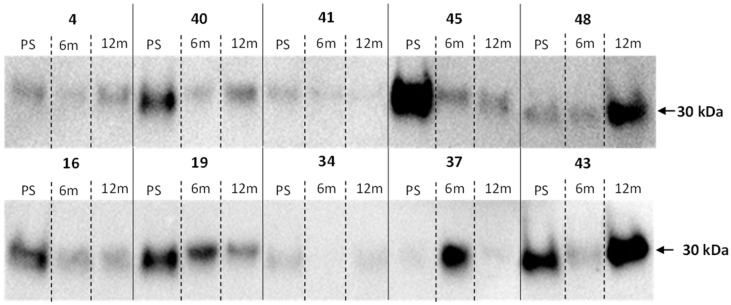
Western blotting of a bariatric patient’s saliva for adiponectin. The numbers correlate with the patient, PS is the pre-Roux-en-Y gastric bypass surgery sample, 6 m is the 6-month follow-up sample, and 12 m is the 12-month follow-up sample.

**Figure 2 ijerph-20-05261-f002:**
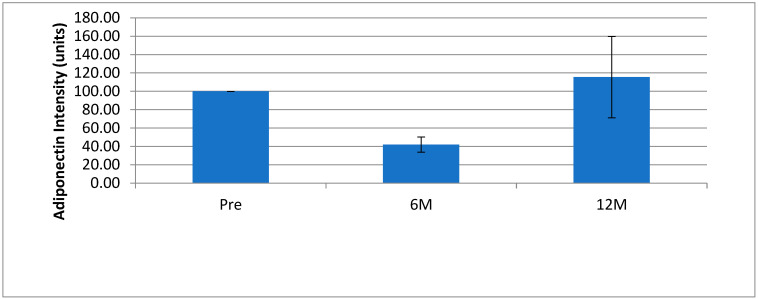
Salivary adiponectin levels showing standard error of bariatric patients before Roux-en-Y (PS) gastric bypass surgery, 6 months (6 m) and 12 months (12 m) after surgery (*n* = 9, sample 37 excluded).

**Figure 3 ijerph-20-05261-f003:**
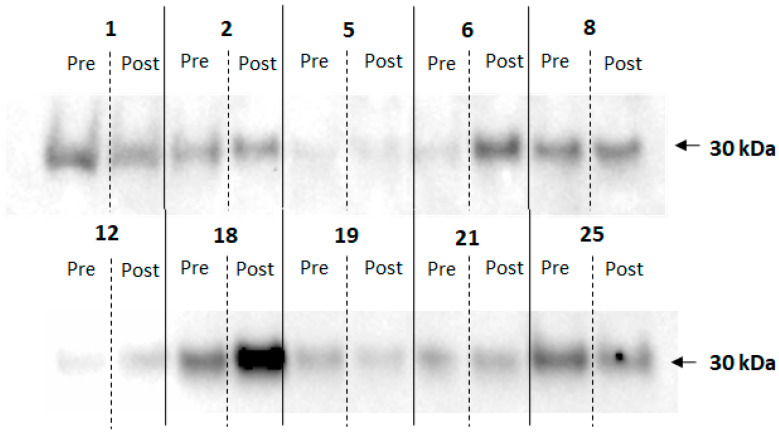
Western blotting of saliva probes from periodontal patients for adiponectin. The numbers correspond to the patient; Pre indicates the saliva sample before and Post after surgery.

**Figure 4 ijerph-20-05261-f004:**
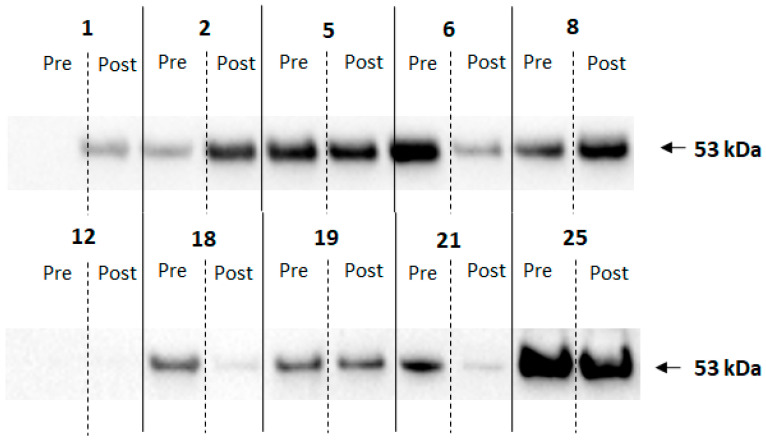
Western blotting of saliva probing from periodontal patients for MMP8. The numbers correspond to the patient; Pre indicates saliva sample and Post after surgery.

**Figure 5 ijerph-20-05261-f005:**
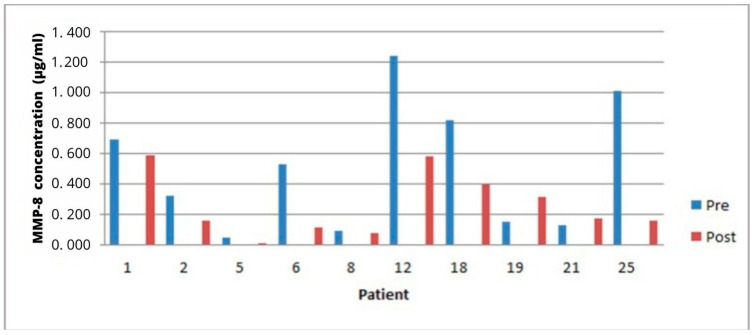
ELISA analysis of the patient’s salivary MMP8 concentration before and after surgery.

**Figure 6 ijerph-20-05261-f006:**
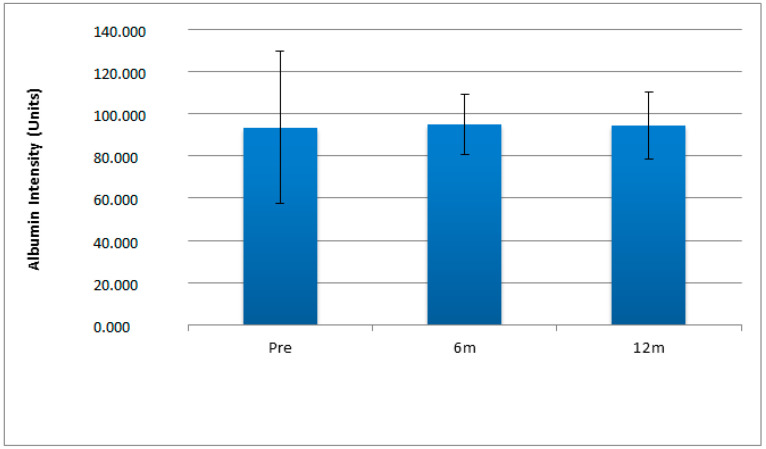
Salivary albumin levels showing standard error of bariatric patients before Roux-en-Y (PS) gastric bypass surgery, 6 months (6 m) and 12 months (12 m) after surgery (n = 10).

**Table 1 ijerph-20-05261-t001:** Mean and standard deviation of the clinical assessments and pocket depth of bariatric patients before and after Roux-en-Y gastric bypass surgery.

Variables	Pre-Surgery	6 Months Post-Surgery	12 Months Post-Surgery
BMI Mean (SD)	50.5 (7.76) ^a^	37.4 (6.07) ^b^	34.0 (5.47) ^b^
Clinical Attachment Loss-CAL Mean (SD)	1.88 (0.53) ^a^	2.17 (0.44) ^b^	2.07 (0.47) ^ab^
Pocket Depth(mm)-PPD Mean (SD)	1.84 (0.52) ^a^	2.09 (0.41) ^a^	2.02 (0.44) ^a^

Different letters in the same line indicate statistically significant differences (ANOVA repeated measures) (*p* < 0.05).

**Table 2 ijerph-20-05261-t002:** Mean and standard deviation of the clinical assessments of periodontal patients before and after periodontal disease treatment by root planning.

Variables	Pre-Treatment Mean ± SD	Post-Treatment Mean ± SD
Gingival Recession (mm)	2.58 ± 0.2 ^a^	3.01 ± 0.29 ^b^
Bleeding on Probing (BoP) (N. Teeth)	18.01 ± 2.07 ^a^	12.5 ± 2.18 ^b^
Pocket Depth (PD) (mm)	3.21 ± 0.22 ^a^	2.90 ± 0.16 ^b^

The lowercase letter corresponds to difference between pre-treatment and post-treatment for periodontal conditions (*p* < 0.05).

## Data Availability

Due to the sensitive nature of the clinical data collected in this study, patients were assured raw data would remain confidential and would not be shared except under specific request.

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
