# Peer review of "Salivary Adiponectin and Albumin Levels on the Gingival Conditions of Patients Undergoing Bariatric Surgery: A Cohort Study"

_ijerph, 2023, doi:10.3390/ijerph20075261_

Round 1

Reviewer 1 Report

Dear authors,

Thank you for submitting your paper to the Q1 IJERPH. I find the topic interesting and I hope that my comments and remarks will be useful in order to improve the quality of the manuscript.

1.     Line 69 – I do agree that MMP-8 is a reliable salivary marker for periodontal inflammation but recent papers point out the significant value of MMP-9 and MMP-13 in quantifying it as well. Please try to emphasise that,  both in the introduction and afterwards in the discussion section.

2.     Line 84- Please motivate why did you choose MMP-8 as a marker of periodontal health.

3.     Did you rely only on BMI or patients underwent Dexa body fat scan? Please clarify and justify this aspect.

4.     Line 113 – How did you determined PD? What type of probes did you use? The same type? Did you use an electronic periodontal probing system? Please clarify and elaborate

5.     What are the limitations of the study?

6.     From my point of view the conclusions section should be rephrased in such a way that it will reflect the practical impact of your findings and the relationship between periodontal disease and obesity.

Kind regards!

Author Response

  1. Line 69 – I do agree that MMP-8 is a reliable salivary marker for periodontal inflammation but recent papers point out the significant value of MMP-9 and MMP-13 in quantifying it as well. Please try to emphasise that,  both in the introduction and afterwards in the discussion section.

We insert in the text:

In the Introduction section: MMP9 is one of the major collagen-degrading enzymes in saliva, which is associated with periodontitis [22] and MMP-13 has also been found to be involved in periodontal tissue destruction and alveolar bone resorption. In fact, MMP-8 (or collagenase 2) is currently one of the most promising biomarkers for periodontitis in oral fluids. [23]

In the Discussion section: Previous study highlighted that MMP9 was associated with obesity and periodontitis [22] and MMP-9 has been shown to be a more sensitive marker for periodontal inflammation. In this way, MMP-8 and MMP-9 are extremely valuable diagnostic tools in treating periodontitis, while MMP-13 affects the activity of osteoclasts and bone resorption, contributing to the destruction of the periodontal tissue.[23]

  1. Line 84- Please motivate why did you choose MMP-8 as a marker of periodontal health.

We insert in the text:

MMP9 is one of the major collagen-degrading enzymes in saliva, which is associated with periodontitis [22] and MMP-13 has also been found to be involved in periodontal tissue destruction and alveolar bone resorption. In fact, MMP-8 (or collagenase 2) is currently one of the most promising biomarkers for periodontitis in oral fluids. [23]

In the methods section: MMP8 was included in this study to investigate the health of periodontium to relate the clinical evaluation.

  1. Did you rely only on BMI or patients underwent Dexa body fat scan? Please clarify and justify this aspect.

In the Methods section: In previous study was demonstrated that anthropometry and DXA were equally useful in obese women, whereas anthropometry had no predictive power and DXA was the only acceptable predictor in obese men. [26] Our sample consisted mostly of women.

Kamel EG, McNeill G, Van Wijk MC. Usefulness of anthropometry and DXA in predicting intra-abdominal fat in obese men and women. Obes Res. 2000 Jan;8(1):36-42.

  1. Line 113 – How did you determined PD? What type of probes did you use? The same type? Did you use an electronic periodontal probing system? Please clarify and elaborate.

The measurements were performed with the use of a standard periodontal clinical probe 15 with a cylindrical shape, 15 mm scale, 1.75° cone taper, and 0.5 mm probe tip diameter (Quinelato, Schobell Industrial Ltda., Rio Claro, São Paulo, Brazil) was used for the clin-ical examination of all present teeth.

The gingival and calculus index were used to measure the presence of bleeding or cal-culus respectively. Bleeding on probing indicates the presence of gingivitis. PPD was measured as the distance between the gingival margin and the bottom of the gingival crevice. CAL was determined by measuring the distance between the cement-enamel junction and the bottom of the gingival crevice. Sites with PPD equal to or greater than 4 mm and with CAL equal to or greater than 3 mm constituted the presence of periodontitis. [28] All patients received professional prophylaxis at baseline.

……….

To dental daily hygiene was used brushing technique was Bass modified, with a soft brush, twice a day, at least with supragingival scaling and polishing have significantly improved the gingival bleeding. 

  1. What are the limitations of the study?

In discussion section: There were some limitations in the present study. First, the sample size was small; however, a causal relationship between weight loss and adiponectin levels of periodontal patients has been established. Second, the population of this study was the convenience sample (non-probabilistic sampling), which may have introduced some selection bias. Finally, the evaluation of saliva and plasma of albumin levels to compare between these biomakers. However, there were several important strengths of our study that are also worth mentioning. This is study to examine the association of salivary albumin and adiponectin with the interaction between bariatric surgery and periodontal disease. In addition, important covariates were measured and controlled in the study analysis.

  1. From my point of view the conclusions section should be rephrased in such a way that it will reflect the practical impact of your findings and the relationship between periodontal disease and obesity.

In the conclusion section: These findings provide insight that the weight loss was not improved the low grade in-flammation in bariatric patients, since the albumin levels was similar between study pe-riods. Periodontitis is an inflammatory disease that is modulated by several factors, among which the adiponectin play an important role for the treatment of periodontal disease in candidates of obesity surgery, while the mechanisms and clinical application require further exploration.

Reviewer 2 Report

All comments are placed in the manuscript:

Page 1

It would be better to write the title: Salivary adiponectin and albumin levels on the gingival conditions of 2 patients undergoing bariatric surgery: a cohort study.

Page 5

According to Song et all. “Gender differences in adiponectin levels and body composition in older adults: Hallym aging study” BMC Geriatr. 2014; 14: 8.

“Plasma adiponectin levels correlated negatively with body fat percentage in older males but not in older females. The differential results between older males and females suggest that certain gender-specific mechanisms may affect the association between adiponectin and age-related body composition changes”.

There is lack information about gender. Maybe it would be worth to divide to male and female?

Page 8:

There is lack information about kidney disease. "Albuminuria is the most prominent symptom of essentially all kidney diseases affecting the kidney glomeruli, and it is independently associated with an increased risk for end-stage renal failure and cardiovascular disease "Cheng and Gerstein.

Page 9:

According to "Comparison of salivary and plasma adiponectin and leptin in patients with metabolic syndrome" there is  correlation between salivary and plasma adiponectin showed significant association (r = .211, p = .018).  It would be worth referring to this article.

Minor changes to table descriptions like:

Salivary albumin..., salivary adiponectin..., bold font etc.

Author Response

Page 1 - It would be better to write the title: Salivary adiponectin and albumin levels on the gingival conditions of 2 patients undergoing bariatric surgery: a cohort study.

We change the title of article.

Page 5

According to Song et all. “Gender differences in adiponectin levels and body composition in older adults: Hallym aging study” BMC Geriatr. 2014; 14: 8.

“Plasma adiponectin levels correlated negatively with body fat percentage in older males but not in older females. The differential results between older males and females suggest that certain gender-specific mechanisms may affect the association between adiponectin and age-related body composition changes”.

A study investigated total levels of adiponectin and its oligomeric profile in saliva from obese subjects compared to age- and sex-matched control subjects. There were no statistical relevant differences of total salivary adiponectin and its oligomers in obese patients compared to healthy controls. [Nigro et al, 2015]

Nigro E, Piombino P, Scudiero O, Monaco ML, Schettino P, Chambery A, Daniele A. Evaluation of salivary adiponectin profile in obese patients. Peptides. 2015 Jan;63:150-5.

There is lack information about gender. Maybe it would be worth to divide to male and female?

The different modulation of adiponectin in saliva and serum of obese patients could be explained assuming that different sources of the adipokines detected in saliva may originate from both local production or may be transported by other biological fluids. [Kardes et al., 2010] Salivary determinations of adiponectin may contribute to the elucidation of the physiology and the role of this adipokinein various conditions, such as obesity, insulin resistance, reproduction, energy balance, and stress response and may represent a driving factor for insulin resistance, local inflammation in patients with obesity and obesity-related diseases. [Nigro et al, 2015] 

Kardes¸ ler L, Buduneli N, Cetinkalp S, Kinane DF. Adipokines and inflammatorymediators after initial periodontal treatment in patients with type 2 diabetesand chronic periodontitis. J Periodontol 2010;81(1):24–33.

Page 8:

There is lack information about kidney disease. "Albuminuria is the most prominent symptom of essentially all kidney diseases affecting the kidney glomeruli, and it is independently associated with an increased risk for end-stage renal failure and cardiovascular disease "Cheng and Gerstein.

A recent systematic review analyzed albumin concentration found that chronic kidney disease patients with periodontitis presented lower albumin levels, indicating that their health was at risk. The authors hypothesized that there might be an inverse relationship between periodontal disease and serum albumin concentration.[Tavares etl., 2022]

Tavares LTR, Saavedra-Silva M, López-Marcos JF, Veiga NJ, Castilho RM, Fernandes GVO. Blood and Salivary Inflammatory Biomarkers Profile in Patients with Chronic Kidney Disease and Periodontal Disease: A Systematic Review. Diseases. 2022 Feb 17;10(1):12.

Page 9:

According to "Comparison of salivary and plasma adiponectin and leptin in patients with metabolic syndrome" there is  correlation between salivary and plasma adiponectin showed significant association (r = .211, p = .018).  It would be worth referring to this article.

 The number of article was insert in the text.

Minor changes to table descriptions like:

Salivary albumin..., salivary adiponectin..., bold font etc.

 This was adjusted.

Reviewer 3 Report

This very interesting article evaluates the possible influence of weight loss on the levels of adiponectin and albumin in the saliva of patients with periodontal disease that undergo bariatric surgery.

After revising the manuscript, my comments are listed as follow:

1)    Please be more precise when explaining what is periodontal disease. The paragraph “Periodontal disease is a chronic disease characterized by pathogenic levels of periodontal pathogens, gram-negative and proteolytic bacteria as a result of inflammatory and immune-mediated tissue changes. Alterations in hormonal balance can alter the microbial homeostasis that maintains healthy gums, leading to the development of periodontal disease.” (lines 64-68) should be reformulated by following updated and important references in the field, for example I suggest to follow this reference related to the periodontal disease definition: Tonetti, M. S., Greenwell, H., & Kornman, K. S. (2018). Staging and grading of periodontitis: Framework and proposal of a new classification and case definition. Journal of periodontology, 89, S159-S172.

2)    Please be careful when discussing about the changes that occurred concerning the periodontal condition. Do you really think that the difference between baseline and 12-month values are clinically relevant? It does not seems to be so (3.21 mm vs 2.90 mm pocket depth). Thus, I suggest to explain it into the section “Discussion”, improving the paragraph “Analysis of the clinical data shows a significant increase in pocket depth and clinical attachment level 6 months after surgery, followed by a significant decrease in 6 month levels 12 months after surgery. This suggests a worsening of the periodontal condition of patients undergoing Roux-en-Y gastric bypass surgery, followed by a return to a healthier periodontal condition,” (lines 269-273) in order to not give to the reader a misleading message.

3)    I suggest to change “clinical indices of gingival health” (line 194) with a more specific term as “clinical periodontal measurements”.

4)    The correct definition of the periodontal probe that was used is “University of North Carolina” (line 110).

5)    Usually, when performing periodontal assessment, by definition, plaque score/index and not calculus is recorded (line 112).

6)    Please explain with details (e.g. which tecnique was explained to the patients) what consist “standard oral hygiene instructions” (line 116).

7)    Please be more precise and consistent when writing your conclusions. This sentence “These findings provide insight into the interrelationships of a systemic anti-inflammatory and pro-inflammatory biomarker in patients with weight loss after surgery.” (lines 322-324) is too generic and does not highlight the findings of this study.

8)    The meaning of the following sentences is not properly addressed. Please, re-write these phrases in a better way:

“This objective study to investigate the influence of weight loss on the levels of adiponec- tin and albumin in the saliva of patients with periodontal disease in patients undergoing bariatric surgery.” (lines 13-15)

“A full-mouth periodontal examination was conducted by trained one examiner to assess probing pocket depth, clinical attachment loss, and bleeding on probing (BOP) and saliva collection (stimulate saliva) were recorded at each visit baseline 6 and 12 months.” (lines 18-21)

“The adiponectin shows the better biomarker for obesity in saliva and it could be down to systemic inflammation post-surgery.” (lines 18-19)

“Since obesity leads to an imbalance in the levels of pro-inflammatory cytokines, affecting the immune response of obese patients.” (lines 73-74)

“Kshirsagar et al. severe periodontitis was associated with low albumin levels in patients with renal failure.” (lines 78-79)

“No significant difference was found between preoperative nor patient salivary albumin levels 6 or 12 months after bariatric surgery (p > 0.05) (Figure 6).” (lines 232-233)

Author Response

This very interesting article evaluates the possible influence of weight loss on the levels of adiponectin and albumin in the saliva of patients with periodontal disease that undergo bariatric surgery.

 After revising the manuscript, my comments are listed as follow:

1)    Please be more precise when explaining what is periodontal disease. The paragraph “Periodontal disease is a chronic disease characterized by pathogenic levels of periodontal pathogens, gram-negative and proteolytic bacteria as a result of inflammatory and immune-mediated tissue changes. Alterations in hormonal balance can alter the microbial homeostasis that maintains healthy gums, leading to the development of periodontal disease.” (lines 64-68) should be reformulated by following updated and important references in the field, for example I suggest to follow this reference related to the periodontal disease definition: Tonetti, M. S., Greenwell, H., & Kornman, K. S. (2018). Staging and grading of periodontitis: Framework and proposal of a new classification and case definition. Journal of periodontology, 89, S159-S172.

 The bacterial biofilm formation initiates gingival inflammation; however, periodontitis initiation and progression depend on dysbiotic ecological changes in the microbiome in response to nutrients from gingival inflammatory and tissue breakdown products that en-rich some species and anti-bacterial mechanisms that attempt to contain the microbial challenge within the gingival sulcus area once inflammation has initiated. [Tonetti et al., 2018]

2)    Please be careful when discussing about the changes that occurred concerning the periodontal condition. Do you really think that the difference between baseline and 12-month values are clinically relevant? It does not seems to be so (3.21 mm vs 2.90 mm pocket depth).

Thus, I suggest to explain it into the section “Discussion”, improving the paragraph “Analysis of the clinical data shows a significant increase in pocket depth and clinical attachment level 6 months after surgery, followed by a significant decrease in 6 month levels 12 months after surgery. This suggests a worsening of the periodontal condition of patients undergoing Roux-en-Y gastric bypass surgery, followed by a return to a healthier periodontal condition,” (lines 269-273) in order to not give to the reader a misleading message.

 The change in periodontal disease between baseline and 12 months was from 3.21 mm to 2.90 mm pocket depth, which is not clinical relevant.

Analysis of the clinical data shows a significant increase in pocket depth and clinical attachment level 6 months after surgery, followed by a decrease from  6 month to 12 months after surgery. The change in periodontal disease between baseline and 12 months was from 3.21 mm to 2.90 mm pocket depth, which is not clinical relevant.

3)    I suggest to change “clinical indices of gingival health” (line 194) with a more specific term as “clinical periodontal measurements”.

 This suggestion was accepted.

4)    The correct definition of the periodontal probe that was used is “University of North Carolina” (line 110).

The measurements were performed with the use of a standard periodontal clinical probe 15 with a cylindrical shape, 15 mm scale, 1.75° cone taper, and 0.5 mm probe tip diameter (Quinelato, Schobell Industrial Ltda., Rio Claro, São Paulo, Brazil) was used for the clinical examination of all present teeth.5)    Usually, when performing periodontal assessment, by definition, plaque score/index and not calculus is recorded (line 112).

 This was a limitation, due to the plaque was not collected by all patients in three periods.

6)    Please explain with details (e.g. which tecnique was explained to the patients) what consist “standard oral hygiene instructions” (line 116).

To dental daily hygiene was used brushing technique was Bass modified, with a soft brush, twice a day, at least with supragingival scaling and polishing have significantly improved the gingival bleeding.

7)    Please be more precise and consistent when writing your conclusions. This sentence “These findings provide insight into the interrelationships of a systemic anti-inflammatory and pro-inflammatory biomarker in patients with weight loss after surgery.” (lines 322-324) is too generic and does not highlight the findings of this study.

– This was re-writed.

These findings provide insight that the weight loss was not improved the low grade inflammation in bariatric patients, since the albumin levels was similar between study periods. Periodontitis is an inflammatory disease that is modulated by several factors, among which the adiponectin play an important role for the treatment of periodontal disease in candidates of obesity surgery, while the mechanisms and clinical application require further exploration.

8)    The meaning of the following sentences is not properly addressed. Please, re-write these phrases in a better way:

 “This objective study to investigate the influence of weight loss on the levels of adiponec- tin and albumin in the saliva of patients with periodontal disease in patients undergoing bariatric surgery.” (lines 13-15)

– This was corrected.

 This study analyzed the salivary proteomics, adiponectin and albumin, related to weight loss and periodontitis in patients undergoing bariatric surgery.

“A full-mouth periodontal examination was conducted by trained one examiner to assess probing pocket depth, clinical attachment loss, and bleeding on probing (BOP) and saliva collection (stimulate saliva) were recorded at each visit baseline 6 and 12 months.” (lines 18-21)

 – This was corrected.

The variables assessed were: probing pocket depth (PPD), clinical attachment loss (CAL), bleeding on probing (BOP), and stimulated whole-mouth saliva.

“The adiponectin shows the better biomarker for obesity in saliva and it could be down to systemic inflammation post-surgery.” (lines 18-19)

 – This was corrected.

The weight loss was not improved the low grade inflammation in bariatric patients, since the albumin levels was similar between periods. Periodontitis is an inflammatory disease that is modulated by several factors, among which the adiponectin play an important role for the treatment of periodontal disease.

“Since obesity leads to an imbalance in the levels of pro-inflammatory cytokines, affecting the immune response of obese patients.” (lines 73-74)

– This was corrected.

 Obesity provides an imbalance in the levels of pro-inflammatory cytokines, impairing the immune response of obese patients. [10]

“Kshirsagar et al. severe periodontitis was associated with low albumin levels in patients with renal failure.” (lines 78-79)

– This was corrected.

 In present study we found no correlation between salivary albumin levels pre-surgery, nor 6 months or 12 months after surgery. Contradictory results have been reported about the relationship between severe periodontitis with low albumin levels. [24]

“No significant difference was found between preoperative nor patient salivary albumin levels 6 or 12 months after bariatric surgery (p > 0.05) (Figure 6).” (lines 232-233)

– This was corrected.

No significant difference was found between the pre-surgery, 6 month post-surgery or 12 month post-surgery bariatric patient salivary albumin levels (p > 0.05) (Figure 6).

Reviewer 4 Report

Firstly, some minor technical issue, from my knowledge the references should be positioned before the full stop.

Line 78 the sentence does not seem to make any sense.

Line 109 says “calibrated dentist” please rephrase so that it uses proper terms.

Line 113 says “were described as 113 previously described at baseline” please correct

Regarding the study, I understand that your study lot features 14 patients, this is not a particularly large lot however you do perform some advanced lab test. Regarding your methods, while periodontal disease may be worse in obese patients your study does not mention oral hygiene in any way. And as numerous studies suggest, hygiene plays a paramount role when it comes to periodontal disease. At least you should have mentioned that all the patients in the study lot had similar oral hygiene, especially that the study lot only includes 14 individuals. This should be cleared up.

While you mention weight loss after surgery, which judging by the table was successful I think the authors could have considered underlining this better in the text. Also, your results show mainly the same adiponectin levels despite the significant weight loss with no improvement in periodontal disease and again not mentioning individual oral health care routine. Otherwise, this study is quite bizarre because in no instance do the authors refer to the patient’s oral hygiene habits or status while putting accent on periodontal disease. For example, it is not mentioned if the patient had gone through any oral procedures beforehand

A welcome addition to the study would have been quantifying blood levels of adiponectin in order to compare to salivary levels as well.

Also, in the end you mention “This is the pioneering study to examine the association of weight loss and albuminuria with the interaction between bariatric surgery and periodontal disease” however when does the study refer to albuminuria? Did you measure albuminuria in these patients?

Author Response

Comments and Suggestions for Authors

Firstly, some minor technical issue, from my knowledge the references should be positioned before the full stop.

Line 78 the sentence does not seem to make any sense.

– This was corrected.

Line 109 says “calibrated dentist” please rephrase so that it uses proper terms.

– This was corrected.

Line 113 says “were described as 113 previously described at baseline” please correct

– This was corrected.

Regarding the study, I understand that your study lot features 14 patients, this is not a particularly large lot however you do perform some advanced lab test. Regarding your methods, while periodontal disease may be worse in obese patients your study does not mention oral hygiene in any way. And as numerous studies suggest, hygiene plays a paramount role when it comes to periodontal disease. At least you should have mentioned that all the patients in the study lot had similar oral hygiene, especially that the study lot only includes 14 individuals. This should be cleared up.

This topic was cleared up in the text.

While you mention weight loss after surgery, which judging by the table was successful I think the authors could have considered underlining this better in the text. Also, your results show mainly the same adiponectin levels despite the significant weight loss with no improvement in periodontal disease and again not mentioning individual oral health care routine. Otherwise, this study is quite bizarre because in no instance do the authors refer to the patient’s oral hygiene habits or status while putting accent on periodontal disease. For example, it is not mentioned if the patient had gone through any oral procedures beforehand

All patients received professional prophylaxis at baseline.

A welcome addition to the study would have been quantifying blood levels of adiponectin in order to compare to salivary levels as well.

This is a limitation of study.

This is study to examine the association of salivary albumin and adiponectin with the interaction between bariatric surgery and periodontal disease. In addition, important covariates were measured and controlled in the study analysis.

Also, in the end you mention “This is the pioneering study to examine the association of weight loss and albuminuria with the interaction between bariatric surgery and periodontal disease” however when does the study refer to albuminuria? Did you measure albuminuria in these patients?

This is the pioneering study to examine the association of weight loss and salivary albuminum and adiponectin with the interaction between bariatric surgery and periodontal disease. Our data highlighted that bariatric and periodontitis patients presenting similar albumin levels, indicating that the weight loss was not improved the low grade inflammation.

Round 2

Reviewer 4 Report

Many of the introduced phrases require English revision such as line 380 to 382, 139 to 142, please reread your manuscript carefully and edit the phrases that have been translated erroneously. 

Perhaps you should edit the title in a way that it reads better in English such as The relation between Salivary adiponectin and albumin levels and the gingival conditions of patients underdoing bariatric surgery .... 

Author Response

Please the attached file.